# Effect of Soil Type and In Vitro Proliferation Conditions on Acclimation and Growth of Willow Shoots Micropropagated in Continuous Immersion Bioreactors

**DOI:** 10.3390/plants12010132

**Published:** 2022-12-27

**Authors:** Carmen Trasar-Cepeda, Conchi Sánchez, Mar Casalderrey, Diana Bello, Jesús María Vielba, Saleta Rico, Anxela Aldrey, Nieves Vidal

**Affiliations:** Misión Biológica de Galicia Sede Santiago de Compostela, Consejo Superior de Investigaciones Científicas, Apdo 122, 15780 Santiago de Compostela, Spain

**Keywords:** liquid medium, photoautotrophic growth, photomixotrophic growth, soil enzyme activities, sucrose, soil organic matter

## Abstract

*Salix viminalis* L. is a species with high capacity for micropropagation and acclimation and could therefore be used to evaluate emergent techniques in the field of plant propagation. The aims of this study were to propagate willow in liquid medium with a continuous immersion system, to explore the application of photoautotrophic conditions and to investigate the adaptation of willow plantlets to different soils that could be used as alternatives to commercial peat. For proliferation, we used 3% sucrose or sugar-free medium, and as substrates, we used commercial peat, a soil from an oak forest with high organic matter content and a crop soil with low organic matter content. The effect of sugar supplementation during proliferation and the soil characteristics during acclimation and growth were evaluated on the basis of aerial and root growth and the hydrolytic and dehydrogenase enzymatic activities of the soils. The results indicate that under photoautotrophic conditions, the supplementation of sucrose during micropropagation did not affect the subsequent growth of the plantlets. All plants acclimated without loss, but the type of soil influenced the height and vigor. Plants produced the highest shoots in peat, whereas the most root development occurred in crop soil. Soil enzyme activities were more influenced by the type of soil than by the presence of plants.

## 1. Introduction

The acclimation process is a major issue for the micropropagation of many plants [1]. Plants are usually micropropagated photomixotrophically, meaning they obtain their energy and biomass from the medium supplied with sugar, grown in small, air-tight vessels with high humidity and low gas exchange. Shoots experience CO_2_ depletion during most of the photoperiod and are exposed to relatively high ethylene concentrations and relatively low photosynthetic photon flux density (PPFD), contributing to disturbances in plant development and photosynthetic performance [2]. After transfer to ex vitro conditions plants have to correct the abnormalities, and aerial and root sections have to adapt to the new environments in the greenhouse or in the field [3,4].

The use of liquid media by continuous (CIS) or temporary immersion (TIS) in bioreactors with forced ventilation has been proposed as a means of improving the physiological status of the explants, enhancing the photosynthetic ability and making them more competent to undergo rooting and acclimation [5,6,7,8,9,10,11]. In our laboratory, we have used CIS to culture *Castanea* spp. [12,13] and TIS to culture *Alnus glutinosa* [14], *Prunus domestica* [15], *Cannabis sativa* [16], *Castanea* spp. [13,17] and *Salix viminalis* [18,19]. *Salix* spp. and hybrids have been micropropagated in semisolid medium [20,21,22,23,24,25,26]. For acclimation, these authors reported a first step in which plantlets were maintained under high humidity conditions followed by a second step in which humidity was gradually decreased to adapt the plantlets to ambient conditions. The length of these phases, as well as the use of an environmentally controlled chamber, phytotron and greenhouse varied among studies [20,21,22,23,24,25,26]. Successful acclimation has been frequently reported, although genotype and a small plant size at the time of ex vitro transfer were reported to decrease survival [21,26].

Within genus *Salix*, *S. viminalis* exhibits a high capacity for micropropagation and acclimation [18] and can therefore be used to evaluate emergent techniques in the field of plant propagation using photoautotrophic growth [19]. Photoautotrophic micropropagation (PAM) is defined as micropropagation without sugar in the culture medium, in which the growth or accumulation of carbohydrates of cultures is fully dependent upon photosynthesis and inorganic nutrient uptake [2]. This technique requires relatively high PPFD and CO_2_ concentrations in the culture environment. We recently developed a protocol to photoautotrophically micropropagate willow shoots by TIS [19]. In that study, we used commercial RITA© bioreactors; their relatively small size (1 L) makes them useful in experiments with a high number of treatments, but they are too small for large-scale propagation. In the present study, we explored the feasibility of culturing willow photoautotrophically using larger containers (10 L) that were operated by continuous immersion.

Roots formed in vitro may not be completely functional [27,28,29,30], and substrate characteristics such as water and oxygen availability, density, porosity and pH can determine plant adaptation to ex vitro conditions. Micropropagated plants are usually transferred to soilless substrates that provide a root environment that is initially free of plant pathogens and has properties that ensure adequate aeration and water and nutrient supply [31]. Peat is the most used substrate constituent in horticulture and micropropagation [31,32,33]. However, peat is a limited resource that is renewable only over an extremely long time, and the extraction of peat bogs negatively impacts the environment, making it necessary to identify more sustainable alternative solutions [31,32,33]. Some studies have investigated the suitability of using mixtures of peat and renewable materials such as rice husk, as well as products from anaerobic digestion of compost and vineyard pruning [33,34,35]. The use of natural soils has also been explored, with the aim of reducing the economic and environmental cost of preparing plants for use in reforestation programs under the stressful environmental conditions typical of degraded areas and promoting early mycorrhization [36,37,38,39].

In the present study, following the first weeks of acclimation in the phytotron, we investigated the growth of willow plantlets in the greenhouse using soil from an oak forest with high organic matter content and a crop soil with low organic matter content as alternatives to peat substrate. We evaluated soil suitability based on aerial and root growth and on the hydrolytic and oxidoreductase (dehydrogenase) enzymatic activities of the soils. Soil enzymes catalyze reactions that transform organic matter and release inorganic nutrients for plant growth and nutrient cycling [40,41,42]. They are easier to measure than other parameters of the soil and are useful biological soil quality indicators [43]. In this study, we investigated the changes in six soil enzyme activities after the willow plantlets grew in peat and crop and forest soils. We analyzed five hydrolytic enzymes (urease, acidic phosphomonoesterase, ß-glucosidase, invertase and arylsulfatase) and dehydrogenase, one of the most studied oxidoreductases, which is very sensitive to changes in soil quality due to management [44]. Hydrolases are associated with carbon, nitrogen, phosphorus and sulfur cycles. Dehydrogenase is an intracellular enzyme that participates in the early stages of organic matter oxidation and is related to soil respiration [45].

The aim of the present study was to develop a protocol for the photoautotrophic micropropagation of willow by continuous immersion in vitro. In addition, we investigated the role of the substrate used in the transfer to ex vitro conditions in the acclimation and growth of shoots cultured with or without sugar, all with the intention of developing protocols applicable to other woody species.

## 2. Results

### 2.1. Effect of Support Material and Sucrose Supplementation on the Micropropagation of Willow Shoots in CIS

In a first experiment, we explored the feasibility of culturing willow shoots in 10 L bioreactors (Figure 1a). Shoots were cultured in CIS in a medium supplemented with 3% sucrose, either directly immersed in liquid medium (Figure 1b) or using a support to maintain the shoot sections in an upright position (in this case, 1 cm^3^ rockwool cubes; Figure 1c).

The explants cultured in the upright position with rockwool cubes grew vigorously and had a multiplication coefficient (MC) of 7 (Figure 2). ANOVA results (Appendix A) indicate that explants grown without support produced a similar number of shoots with heights similar to those grown with rockwool cubes (*p* = 0.940 and 0.311, respectively). However, without support, a high frequency of hyperhydricity was observed. 

The percentage of hyperhydric shoots was less than 2% in containers with cubes, whereas in the bioreactor without support, more than 70% of the new shoots were affected, leading to a significant reduction in the multiplication coefficient (MC; 3.9 compared with 6.9, *p* < 0.001). In a subsequent trial, we tested the effect of gas exchange, inserting willow explants on vessels with rockwool cubes and without aeration. We obtained more new shoots (6.4 per initial explant), but more than 90% of them were hyperhydric, with MC reaching only 1.4. Aeration was always applied in all the experiments thereafter.

In next experiment, we compared two sucrose supplementations (S0 and S3, corresponding to 0 and 3% sucrose, respectively) and two types of support: rockwool cubes and plastic boxes with holes, which were pipette tip boxes (Figure 1d). Sucrose did not affect the number of shoots (NS) or the shoot length (SL), regardless the support (*p* = 0.202 and 0.216, respectively) or MC for explants cultured with cubes, whereas those cultured in boxes proliferated more without sucrose (Figure 3, Appendix A). The choice of support had a significant effect (*p* = 0.029 for NS and *p*< 0.001 for SL and MC), and higher values were obtained when plastic boxes were used. In contrast, the photosynthetic pigments were more affected by the sucrose and the interaction of sucrose × support than by the support itself (Figure 4, Appendix A). For sucrose, *p* values were lower than 0.001 for all parameters, similarly to the interaction of sucrose × support (lower than 0.006), whereas for support, *p* values of 0.851, 0.446 and 0.736 were obtained for chlorophyll a and b and total carotenoids, respectively. Among the shoots cultured with plastic boxes, more pigments were observed when shoots were cultured in S0, whereas no significant differences were detected between S0 and S3 in the case of the cubes.

Explants cultured in plastic boxes produced more and longer shoots than those cultured with rockwool cubes, irrespective of the sucrose treatment (Figure 3 and Figure 5a–d). However, the roots spontaneously formed by willow shoots grew through the holes of the plastic boxes, creating an intricate root net that made it difficult to extract the shoots and handle them in the laminar flow cabinet during subculturing (Figure 5e).

Therefore, we used rockwool cubes for the experiments with the intention of producing plant material for acclimation studies. Rockwool-supported explants treated with S0 produced similar aerial growth to those treated with S3 (Figure 3). Shoot biomass was also similar (230 mg DW/explant for S3 and 270 for S0). All shoots formed roots spontaneously during the proliferation stage, although shoots cultured with S3 formed approximately twice as much root weight as shoots grown with S0 (410 mg DW/explant and 192 mg DW/explant, respectively).

### 2.2. Effect of Soil Type on the Acclimation of Shoots Micropropagated with or without Sucrose

The first step of the acclimation process was carried out in a phytotron. Shoots were removed from bioreactors and measured before being planted in pots (Figure 6a).

Shoot length averaged 6.8 and 6.9 cm for S3 and S0 treatments, respectively, before planting, but in pots, the aerial height was reduced to 5.2 and 5.6 cm, respectively (*p* = 0.499; Figure 7, Appendix A) as a result of some stems being covered by growing medium. 

Basal stem diameter averaged 1.3 mm in both treatments. After 4 weeks in the phytotron, the height of plants cultured with S3 was significantly higher than those cultured without sugar (45.5 and 38.9 cm, respectively; *p* = 0.001; Appendix A). Growth slowed down during the first weeks after transplantation to new pots in the greenhouse (Figure 6b and Figure 7). Upon completion of the experiment, S3 plantlets averaged 62.2 cm height and S0 plantlets averaged 58.2 cm height (*p* = 0.217). The soil type had a significant effect on plant height (*p* < 0.001), and the three groups (peat, forest and crop soil) showed significant differences between them (Appendix A). No interaction between soil and sucrose supplementation was detected (*p* = 0.745). The highest aerial growth was obtained in peat (67 cm), followed by crop soil (57 cm) and forest soil (49 cm) (Figure 6c and Figure 7). The basal diameter averaged the same value for the two sucrose treatments (4.6 mm) and was influenced by soil type (*p* = 0.002; Appendix A), with values ranging between 5.6 (PS0) and 4.0 mm (CS3).

Figure 6c–i and Figure 8 show the biomass of the aerial and root zones upon completion of the experiment. The aerial biomass followed a similar trend to that of the shoot heights shown in Figure 7, whereas roots developed more on plantlets grown in crop soil. Two-way ANOVA (Appendix A) indicated that sucrose, soil and sucrose × soil significantly affected shoot growth (*p* = 0.008, < 0.001 and 0.029, respectively), whereas root biomass was only affected by the soil type (*p* = 0.003) and was not affected by sucrose or sucrose × soil (*p* = 0.662 and 0.317, respectively).

Photosynthetic pigments (chlorophylls a and b and total carotenoids) of plantlets after acclimation at 6 weeks are shown in Figure 9, and the two-way ANOVA results are presented in Appendix A.

In this case, both sucrose and soil type significantly affected the pigment concentration (*p* < 0.001 in all cases), and no interaction between sucrose and soil was observed (*p* = 0.735, 0.652 and 0.843 for chlorophyll a and b and carotenoids, respectively). More pigments were formed in leaves of plantlets that had been cultured with S3 and had been planted in peat, whereas in forest and crop soil, similar values were obtained (Figure 9, Appendix A). 

### 2.3. Main General Properties of the Soils Used in the Study

The main general properties of the soils are shown in Table 1. As expected, commercial peat contained more total C and N and available nutrients (inorganic P and N) than the forest soil (rich in organic matter) and the crop soil. The latter soil had less organic matter content and fewer nutrients than peat and forest soil.

### 2.4. Enzymatic Activities of the Soil

The activities of six soil enzymes were analyzed using bulk soil samples after being in contact with willow plant roots for 6 weeks (Table 2). The results were more influenced by the type of soil than by the presence of plants, as reflected by the values recorded in soils with plants, very similar to the control pots without plants.

Statistical analysis was performed to compare the enzymatic activities in soils with plants that had been micropropagated with and without sucrose (Appendix A) and indicated that this parameter only caused a significant effect in two of the six enzymes, dehydrogenase (*p* = 0.003) and phosphomonoesterase (*p* = 0.034), with higher values observed in plants micropropagated without sucrose. The soil type effect was highly significant in all enzymes. In the cases of urease, invertase and arylsulfatase, the highest activity was observed in the forest soil, followed by crop and peat soils. With dehydrogenase and ß-glucosidase, peat showed the highest activity, followed by forest soil, whereas phosphomonoesterase showed the highest activity in forest soil, and peat exhibited only slightly more activity than crop soil.

## 3. Discussion

The application of continuous immersion systems to woody plant micropropagation has not reached the same level of acceptance as temporary immersion systems [11]. CIS has been used to culture a wide range of herbaceous plants, such as *Solanum tuberosa* [46], *Spathiphyllum cannifolium* [47], *Alocasia amazonica* [48], *Cymbidium sinense* [49], *Philodendron bipinnatifidum* [50] and *Vanilla planifolia* [51], and well as some woody plants, such as *Vaccinium* spp. [52] and *Castanea* spp. [12,13], but can cause hyperhydricity, as reported in *Malus* M9 EMLA [53,54], *Lessertia frutescens* [55] and *Eucalyptus camaldulensis* [56].

In the present study, we successfully propagated willow shoots by CIS in sugar-free medium using 10 L vessels aerated with CO_2_-enriched air and illuminated with white LEDs, providing a PPFD of 150 μmol m^−2^ s^−1^. We previously used these conditions to achieve the photoautotrophic micropropagation of willow by TIS [19]. In that study, we did not use any support for plant material, and hyperhydricity was not detected, whereas in the current report, we needed a support to maintain the shoots in an upright position to avoid that disorder. Continuous immersion of the whole explant usually causes hyperhydricity, as oxygen concentration is often insufficient to meet the respiratory requirements of submerged tissues [57]. To obtain vigorous willow shoots by CIS, it was necessary to use a support and forced ventilation.

Explants were inserted in holes in recycled pipette tip boxes, which proved to be both a cheap and reusable support. With willow, they provided excellent results in terms of shoot length and multiplication coefficients, but rapid root growth through holes and gaps created a root mesh that was a problem during subcultures. Shukla et al. [58] reported the use of two plastic comb-shaped pieces that were placed at right angles, creating a surface with holes to insert the explants; they claimed their design can be pulled apart in such a way that allows the plants to be removed from the bioreactor without damaging the roots. These authors evaluated its suitability for rooting woody plants using CIS without aeration or a rocker-based TIS. In the present study, we used rockwool cubes that had achieved good results for micropropagation of chestnut by TIS and CIS [12,13,17]. Without reaching the plant performance levels of plastic boxes, cubes provided a good proliferation response and did not cause handling problems, enabling us to obtain a high number of homogeneous, good-quality plantlets for acclimation experiments.

All plants cultured with S3 and S0 survived after ex vitro transfer to the phytotron and to greenhouse conditions, indicating that willow shoots can be propagated photoautotrophically by CIS. We did not observe significant differences in shoot growth and development among the explants cultured with cubes with or without sucrose. However, S0 treatments negatively affected root biomass production in bioreactors. We previously observed a sharp decrease in the root biomass of willow shoots grown in RITA© bioreactors without sucrose [19]. In that study, we successfully acclimated the plantlets that had been micropropagated photoautotrophically but did not record the root biomass after planting in pots. In the present study, we observed that after ten weeks of acclimation and plant growth, root biomass was only affected by the soil type and not by the previous sucrose treatment, confirming the suitability of PAM for mass production of this species.

It has been claimed that shoots cultured by PAM show enhanced photosynthetic competence and produce more growth [2,59]. In the present study, we quantified the photosynthetic pigments of willow shoots during the micropropagation and the acclimation stages. In both cases, the highest values of chlorophylls correlated directly with the best treatment in terms of shoot growth but, for the remaining treatments, did not follow the same trend. Photosynthetic performance and growth are complex traits influenced by many factors. A lack of correlation between photosynthetic pigments and growth has been reported previously for willow [19] and for other plants, such as myrtle [60], chestnut [61], apple [62], tobacco, potato, strawberry, rapeseed [59] and Vernonia [63].

Plantlets grew successfully in the three soils, despite their different nutrient content. Peat, the richest substrate, produced higher shoots than the other soils, whereas the most root development was obtained in crop soil, which had the lowest nutrient content. Plantlets grown in a forest soil did not show an intermediate response as expected, which suggests that other factors may regulate plant growth. Crop soil (with low nutrient content) resulted in relatively good growth of the aerial part and good development of the root system; this could be advantageous for the subsequent transplanting of willows to field conditions. Soils present in areas for restoration are usually not as fertile as the other substrates used in this study (peat and forest soil), and plants cultured in the crop soil seemed to be well-adapted to these conditions.

To investigate the interaction between the willow plants and the soils, we quantified the changes produced by plant roots in six soil enzymes. Urease participates during the nitrogen cycle by catalyzing the hydrolysis of urea to form ammonia and CO_2_. It is widely distributed and present in microorganisms, as well as animal and plant cells. Phosphomonoesterase contributes to the mineralization of organic phosphorus, facilitating uptake by microorganisms and plants. β-glucosidase provides an early indication of changes in cellulose-based organic matter by hydrolyzing the cellobiose residue to glucose [44] and therefore supplying energy for soil microorganisms [64]. Invertase is found in microorganisms and radical exudates; it hydrolyzes fructofuranosides such as sucrose, producing glucose and fructose. Arylsulfatase, which is produced by microorganisms in the rhizosphere, catalyzes the hydrolysis of sulfate esters, producing phenols and SO_4_^2−^, which are essential for the nutrition of plants, fungi and bacteria [65], as well as dehydrogenase, which is an intracellular enzyme involved in organic matter oxidation, providing information about microbial activity and oxidative activity of soils [66]. However, the enzymatic activities changed relatively little with the presence of plants, and the differences in activity were mainly related to the type of soil. In the present study, we examined enzymatic activities using bulk soil samples. The lack of evidence of significant differences between the enzymatic activity of the soil with and without plants suggests that the main influence of plants and plant roots occurred in the rhizosphere soil, and it is possible that more than six weeks is required to detect their influence on the bulk soil.

## 4. Materials and Methods

### 4.1. Plant Material and Micropropagation in Liquid Medium

*Salix viminalis* shoot cultures were previously established in vitro from a mature tree [18]. For continuous immersion (CIS) experiments, willow apical and basal sections (25 mm) were cultured in liquid medium (LM) in “in-house” 10 L bioreactors prepared as described by Cuenca et al. [12] for chestnut and designated as C10. Each vessel contained 40 explants and 1000 mL of LM. The LM consisted of MS (Murashige and Skoog) [67] salt and a vitamin mixture with half-strength nitrates (MS-½N) supplemented with 0.22 µM of BA and 0 or 3% sucrose. The medium was adjusted to pH 5.7 before being autoclaved at 120 °C for 20 min. The explants were cultured in an experimental unit [13] designated as PAM, where the cultures grew under high PPFD (150 μmol m^−2^ s^−1^), and CO_2_-enriched air (≈2000 ppm) was injected into the bioreactors for 1 min every hour through 0.2 µm air filters. Cultures were incubated under a 16 h photoperiod at 25 °C light/20 °C dark. After 8 weeks of culture, morphological data were recorded, and shoots were used for a new cycle of proliferation or for acclimation and biochemical analysis.

In the first experiment of application of CIS to willow cultures, we investigated the effect of using support material, i.e., 1 cm^3^ rockwool cubes (Grodan, Roermond, The Netherlands), on the growth of explants cultured with 3% sucrose. In the second experiment, we studied the effect of forced ventilation on the explants grown with rockwool cubes, and in the third experiment, we compared the effect of two parameters: (i) sucrose supplementation (3% under photomixotrophic conditions or 0% under photoautotrophic conditions) and (ii) support material (plastic pipette tips boxes or rockwool cubes). The following responses were recorded: (a) the number of shoots longer than 1.5 cm produced by each explant (NS), (b) the percentage of hyperhydric shoots, (c) the multiplication coefficient (MC) calculated as the number of new segments of 1.5 cm with at least one node obtained from each initial explant, (d) the length of the longest shoot per explant (SL), (e) the biomass of shoots and roots and (f) the level of photosynthetic pigments. For quantitation of pigments, the uppermost two expanded leaves of six explants per vessel were measured and weighed, and the samples were submerged in tubes with dimethylformamide and incubated for 24 h in darkness at 20 °C. Chlorophylls a and b and total carotenoids were quantified spectrophotometrically following the method described by Wellburn [68].

### 4.2. Acclimation and Effect of Soil Type

Rooted shoots obtained by CIS under photomixotrophic and photoautotrophic conditions using rockwool cubes as support material were selected for evaluation of different soils during the acclimation step. Uniform shoots with approximately 70 mm height were chosen, and their roots were cut uniformly to 30 mm length from the base of the shoots and planted in commercial peat for a first acclimation step in 300 mL pots. The pots were transferred to a controlled environmental chamber (Fitotron SGC066, Sanyo Gallencamp PLC) with a photoperiod of 16 h light: 8 h dark, a photosynthetic photon flux of 240–250 µmol m^−2^ s^−1^, a temperature of 25 °C (day) and 20 °C (night) and relative humidity of 85%. The plantlets were sprayed with water daily, watered twice a week and measured at 2-week intervals. Four weeks after transplanting, the aerial and root sections were measured, and roots were carefully washed to eliminate the peat before being transferred to new pots (total volume, 1.3 L; substrate, 1 L) in the greenhouse. Initially, plants were watered to container capacity and, later, watered twice a week by basal immersion for 1–2 h. Three substrates were used: (i) commercial peat, (ii) soil from an oak forest (forest soil) and (iii) an agricultural soil (crop soil), for a total of six groups of four plants (two micropropagation systems and three soils). For each substrate, a control pot without plants was included and subjected to the same conditions as the plants. The main general properties and the enzymatic activities were analyzed before planting, and the soil enzymatic activities were quantified again after six weeks of cultivation. At that time plants were removed from the soil and the following growth responses were assessed: (a) plant and root height, (b) diameter at the base, (c) plant and root biomass and (d) photosynthetic pigments.

### 4.3. Soil Analysis

The total organic C content (wet oxidation with potassium dichromate in strongly acidic medium), the total N content (Kjeldahl digestion) and the pH in water and in 1 M KCl (soil:solution ratio of 1:2.5) were quantified according to Guitián Ojea and Carballas Fernández [69]. Total soil P was determined using 0.2 g of soil digested in a 3:1 mixture of concentrated HNO_3_:HCl, to which 3 mL H_2_O_2_ was added, in Teflon PFA vessels in a laboratory microwave system (Ethos EASY; Milestone, Sorisole, Italy). The content of P in the digestates were determined colorimetrically using the molybdenum blue method [70] with ascorbic acid as a reducing agent. The soluble C content was determined after extraction with hot water (80 °C, 16 h) in a shaking water bath. The extract was centrifuged (3200× *g*, 20 min) and filtered through a 0.45 µm membrane filter [71]. Aliquots of the filtrate were dried at 60 °C, and the total C content was measured in the dried extracts by oxidation with dichromate in acidic medium [69]. The results are expressed in mg C kg^−1^. Inorganic N was extracted with 2 M KCl (1:5 soil:solution ratio), and ammoniacal N and total inorganic N were determined in the extracts by Kjeldahl distillation [72]. The NO_3_^−^-N content of the extracts was calculated as the difference between total inorganic N and NH_4_^+^-N content. The results are expressed in mg N kg^−1^. Available P forms (total, inorganic and organic) were extracted with 0.5 M NaHCO_3_ at pH 8.5 (1:50 soil:solution ratio) and a shaking period of 16 h [73]. Total P in the extracts was determined after digestion of an aliquot of the extract with KMnO_4_ in acidic medium [74]. The extracted organic matter was flocculated with H_2_SO_4_ to pH 1.5. After 16 h the sample was centrifuged for 15 min at 5000 rpm before the inorganic P was determined in the extracts by the method of Murphy and Riley [70] with ascorbic acid as a reducing agent. The organic P was estimated as the difference between total and inorganic P. The results are expressed in mg P kg^−1^.

### 4.4. Determination of Enzyme Activities

Dehydrogenase activity was determined with 0.5% iodonitrotetrazolium violet (INT) as a substrate and incubated with 1 M TRIS-HCl buffer (pH 7.5) for 1 h. The produced iodonitrotetrazolium formazan (INTF) was extracted with a 1:1 (v:v) mixture of ethanol and dimethylformamide and measured spectrophotometrically at 490 nm [75]. Activity was quantified with reference to a calibration curve constructed using INTF standards incubated with soil [75] under the same conditions described above and expressed in µmol INTF g^−1^ h^−1^. The activity of urease was determined as described by Nannipieri et al. [76]. Urease activity was determined by incubating the samples with 1065.6 mM urea as a substrate for 1.5 h in 0.2 M phosphate buffer (pH 8.0; due to the strong buffering capacity of the soils, the pH of the phosphate buffer was 8.0 to ensure that the pH of the reaction mixture was near pH 7.0, which is the optimum for urease activity) and measuring the released NH_4_^+^ with an ammonia electrode. The activity is expressed in µmol NH_3_ g^−1^ h^−1^. Invertase activity was determined by incubating the samples with 35.06 mM saccharose in 2 M acetate buffer (pH 5.5) for 3 h and measuring the released reducing sugars following the method of Schinner and von Mersi [77]. The invertase activity is expressed as µmol glucose g^−1^ h^−1^. Acid phosphomonoesterase activity was determined at pH 5.0, with *p*-nitrophenyl phosphate 16 mM as substrate following the method of Tabatabai and Bremner [78], with 16 mM p-nitrophenyl phosphate as substrate but with some modifications. Modified universal buffer was used to maintain the pH of the reaction mixture as described by Trasar-Cepeda et al. [79]. After 30 min incubation, 2 M CaCl_2_ was added (to prevent dispersion of soil colloids and to avoid the brown coloration caused by organic matter), and the released *p*-nitrophenol was extracted with 0.2 M NaOH [80]. The enzymatic activity was quantified with reference to calibration curves corresponding to *p*-nitrophenol standards incubated with each soil under the same conditions as for the samples [80,81]. ß-glucosidase activity was determined as described for acid phosphomonoesterase activity, except that the substrate was *p*-nitrophenyl-ß-glucopyranoside 25 mM, the incubation time was 1 h and the released *p*-nitrophenol was extracted with THAM-NaOH 0.1 M of pH 12 [82]. Arylsulfatase activity was determined by incubating the samples with 5 mM p-nitrophenyl sulfate as a substrate with 0.5 M acetate buffer (pH 5.8) for 1 h [65]; however, after incubation, 2 M CaCl_2_ and NaOH 0.2 M were used instead of CaCl_2_ and NaOH 0.5 M. The enzymatic activity was quantified with reference to calibration curves corresponding to *p*-nitrophenol standards incubated with each soil under the same conditions as for the samples [80,81]. For the latter three enzymes, the enzymatic activity is expressed in µmol *p*-nitrophenol g^−1^ h^−1^. All data were determined in triplicate, and the mean values are expressed on an oven-dried (105 °C, 24 h) soil weight basis.

### 4.5. Data Recording and Statistical Analysis

For plant growth parameters, data were collected from 60 explants per treatment in micropropagation experiments and from 4 plantlets per treatment in acclimation experiments. For pigment analysis, data correspond to 12 independent samples per treatment. For soil analysis, data were collected from four pots per treatment. The data were analyzed by Levene’s test (to verify the homogeneity of variance) and the Shapiro–Wilk test of normality. The data were then subjected to analysis of variance (ANOVA) followed by comparison of group means (Tukey-b test) or to Welch ANOVA followed by Games–Howell post hoc comparison (when heteroscedasticity was detected). When an interaction between two factors was indicated by the two-way ANOVA, Bonferroni’s adjustment was applied to detect simple main effects. Statistical analyses were performed using SPSS 28.0 (IBM).

## 5. Conclusions

To the best of our knowledge, this is the first study demonstrating the feasibility of micropropagation of *Salix viminalis* shoots in large bioreactors by continuous immersion, both photomixotrophically and photoautotrophically (without sugar). Plantlets acclimated successfully, irrespective of the presence of sugar in the micropropagation medium. All plants grew successfully after transferring to three soils with different nutrient contents, which could facilitate transplanting to field conditions with infertile or degraded soils. Future research will be aimed at developing sustainable supports for the micropropagation step and the first few weeks of ex vitro growth.

## Figures and Tables

**Figure 1 plants-12-00132-f001:**
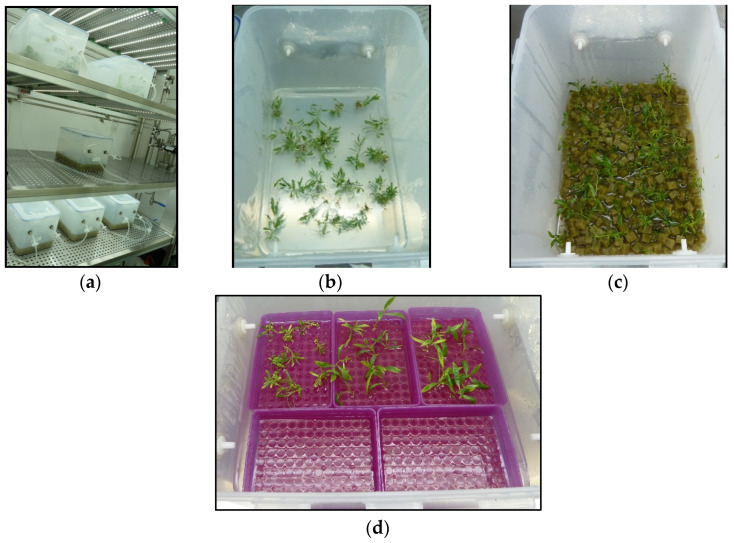
Willow explants micropropagated in continuous immersion bioreactors with different substrates. (**a**) Ten-liter bioreactors used for proliferation. (**b**) Explants cultured without support. (**c**) Explants inserted between 1 cm^3^ rockwool cubes. (**d**) Explants placed on plastic boxes with holes.

**Figure 2 plants-12-00132-f002:**
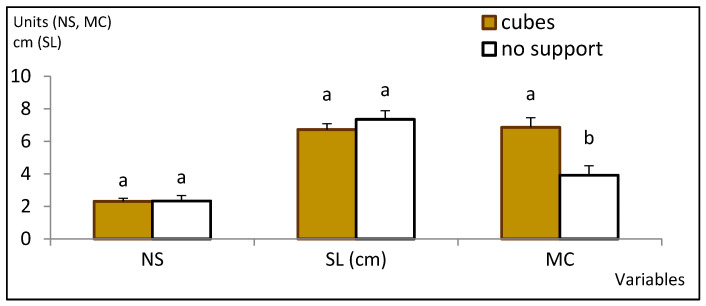
Effect of support material on proliferation rates of willow shoots grown in continuous immersion bioreactors. Shoots were grown in MS medium with half-strength nitrates, 0.22 µM BA and 3% sucrose for 8 weeks. Values are presented as mean ± standard error (3 replicates with 20 explants each). For each variable, different letters indicate significant differences at *p* < 0.05. NS. number of shoots; SL, length of the longest shoot; MC, multiplication coefficient.

**Figure 3 plants-12-00132-f003:**
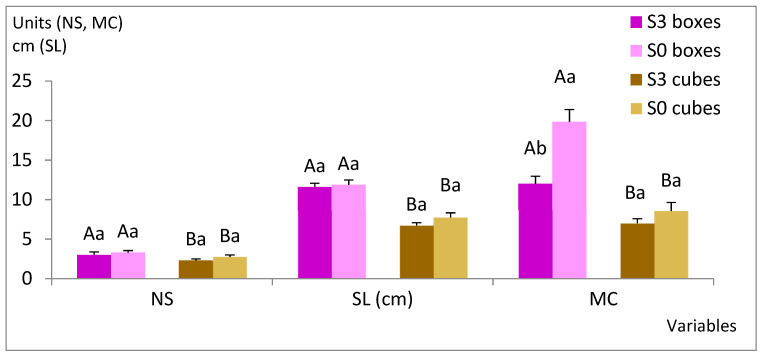
Effect of support material (rockwool and plastic boxes) and sucrose supplementation (S0 and S3, corresponding to 0 and 3% sucrose, respectively) on proliferation rates of willow shoots grown in continuous immersion bioreactors. Shoots were grown in MS medium with half-strength nitrates and 0.22 µM BA for 8 weeks. Values are presented as mean ± standard error (3 replicates with 20 explants each). For each variable, different uppercase letters indicate significant differences in relation to the support, and different lowercase letters indicate significant differences relative to the sucrose supplementation (*p* < 0.05). Due to the significant interaction found in MC, Bonferroni’s adjustment was applied to detect simple main effects. NS, number of shoots; SL, length of the longest shoot; MC, multiplication coefficient.

**Figure 4 plants-12-00132-f004:**
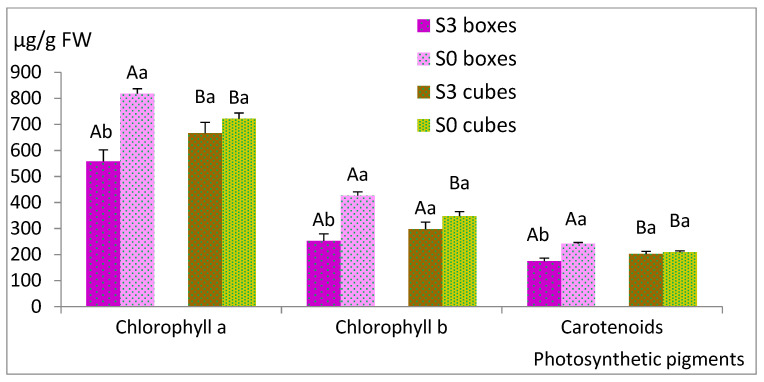
Effect of support material (rockwool and plastic boxes) and sucrose supplementation (S0 and S3, corresponding to 0 and 3% sucrose, respectively) on pigment content of willow shoots grown in continuous immersion bioreactors. Shoots were grown in MS medium with half-strength nitrates and 0.22 µM BA for 8 weeks. Values are presented as mean ± standard error of 12 explants. For each variable, different uppercase letters indicate significant differences in relation to the support, and different lowercase letters indicate significant differences relative to the sucrose supplementation (*p* < 0.05). Due to the significant interaction found in for all parameters, Bonferroni’s adjustment was applied to detect simple main effects.

**Figure 5 plants-12-00132-f005:**
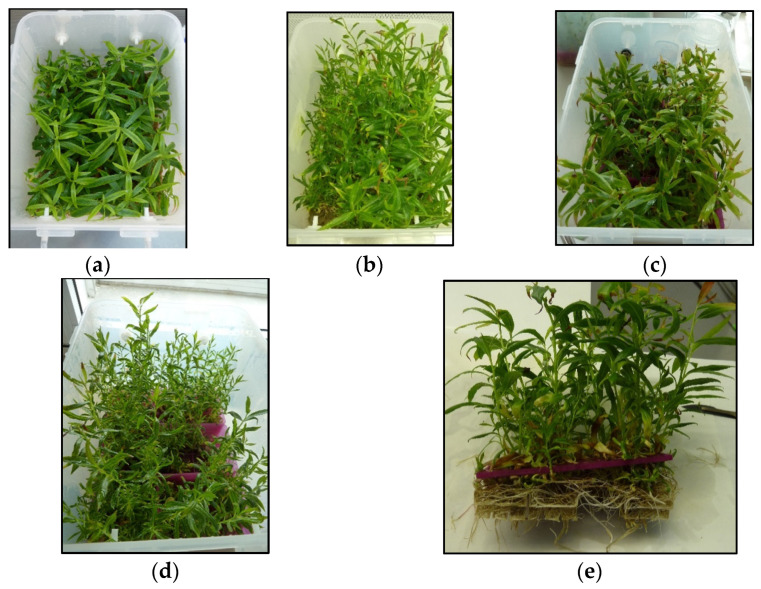
Willow explants after 8 weeks of culture in continuous immersion bioreactors with different substrates. (**a**,**b**) Explants inserted between 1 cm^3^ rockwool cubes and cultured with 3% sucrose (**a**) or without sucrose (**b**). (**c**,**d**) Bioreactors with explants cultured on plastic boxes with 3% sucrose (**c**) or without sucrose (**d**). (**e**) Detail of explants cultured on plastic boxes at the time of subculturing.

**Figure 6 plants-12-00132-f006:**
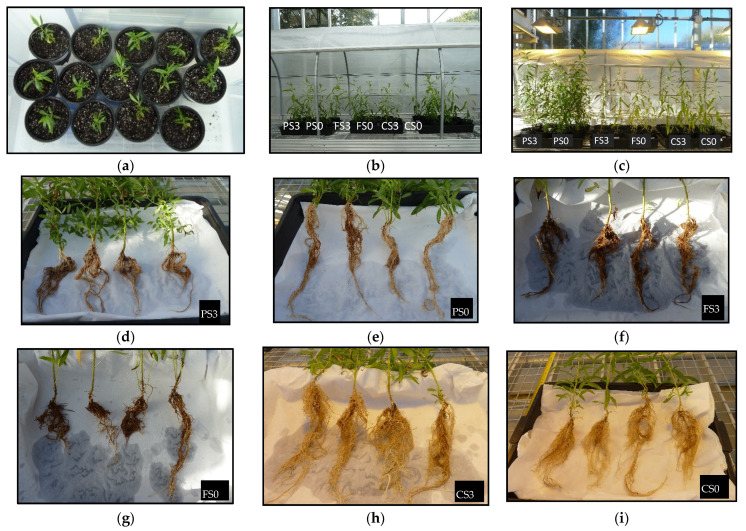
Acclimation of willow plantlets. (**a**–**c**) Plantlets at the time of transfer to phytotron (**a**), to the greenhouse (**b**) and 6 weeks afterwards (**c**). (**d**–**i**) Roots of plants grown in different soils upon completion of the experiment. PS0, PS3: plants grown in peat and micropropagated without sucrose (PS0) or with 3% sucrose (PS3); FS0, FS3: plants grown in a forest soil and micropropagated without sucrose (FS0) or with 3% sucrose (FS3); CS0, CS3: plants grown in a crop soil and micropropagated without sucrose (CS0) or with 3% sucrose (CS3).

**Figure 7 plants-12-00132-f007:**
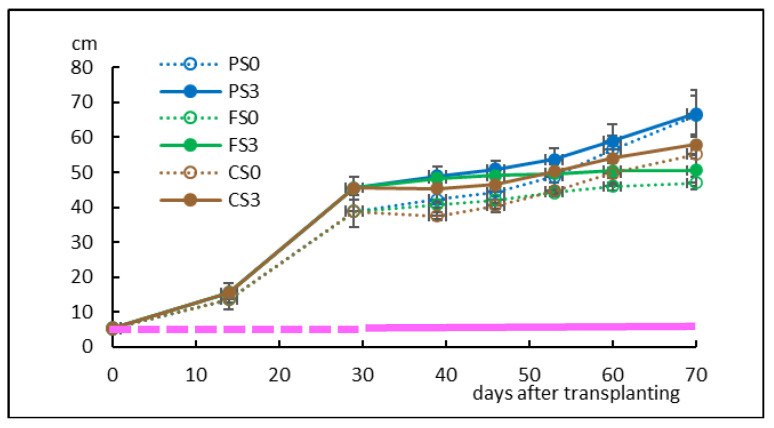
Shoot height of willow plantlets cultured for 4 weeks in the phytotron and for 6 weeks in the greenhouse in three types of soil. PS0, PS3: plants grown in peat and micropropagated without sucrose (PS0) or with 3% sucrose (PS3); FS0, FS3: plants grown in a forest soil and micropropagated without sucrose (FS0) or with 3% sucrose (FS3); CS0, CS3: plants grown in a crop soil and micropropagated without sucrose (CS0) or with 3% sucrose (CS3). Pink dashed line corresponds to the phytotron stage, and pink solid line corresponds to the greenhouse stage. Values are presented as means ± standard error of 14 plants (phytotron stage) and 4 plants (greenhouse stage) per treatment.

**Figure 8 plants-12-00132-f008:**
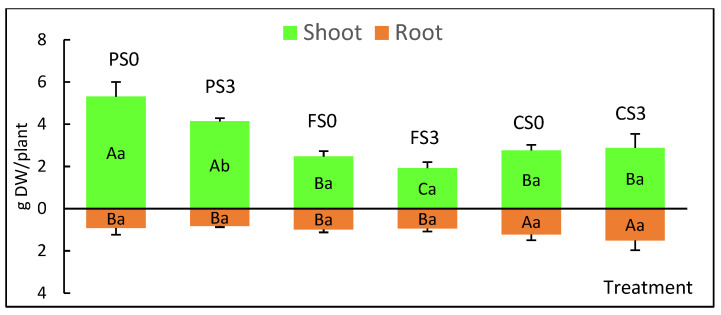
Aerial and root biomass of willow plantlets cultured for 4 weeks in the phytotron and 6 weeks in the greenhouse in three types of soil. PS0, PS3: plants grown in peat and micropropagated without sucrose (PS0) or with 3% sucrose (PS3); FS0, FS3: plants grown in a forest soil and micropropagated without sucrose (FS0) or with 3% sucrose (FS3); CS0, CS3: plants grown in a crop soil and micropropagated without sucrose (CS0) or with 3% sucrose (CS3). Values are presented as means ± standard error of four plants per treatment. For each variable, different uppercase letters indicate significant differences in relation to the soil, and different lowercase letters indicate significant differences relative to the sucrose supplementation (*p* < 0.05). Due to the significant interaction found in aerial growth, Bonferroni’s adjustment was applied to detect simple main effects.

**Figure 9 plants-12-00132-f009:**
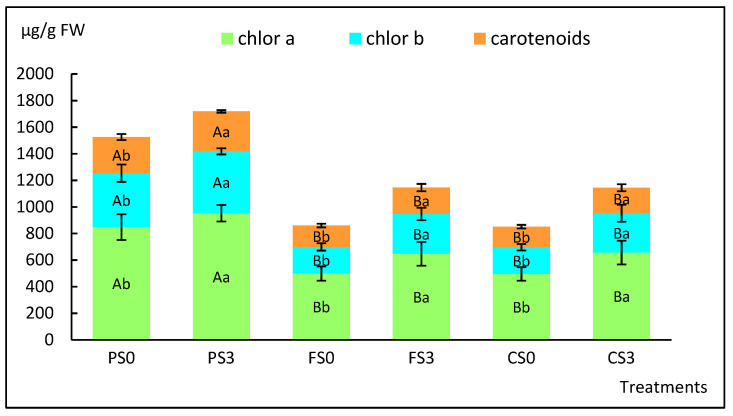
Photosynthetic pigments of willow plantlets cultured for 4 weeks in the phytotron and 6 weeks in the greenhouse in three types of soil. PS0, PS3: plants grown in peat and micropropagated without sucrose (PS0) or with 3% sucrose (PS3); FS0, FS3: plants grown in a forest soil and micropropagated without sucrose (FS0) or with 3% sucrose (FS3); CS0, CS3: plants grown in a crop soil and micropropagated without sucrose (CS0) or with 3% sucrose (CS3). Values are presented as means ± standard error of 12 leaves per treatment. For each variable, different uppercase letters indicate significant differences in relation to the soil, and different lowercase letters indicate significant differences relative to the sucrose supplementation (*p* < 0.05).

**Table 1 plants-12-00132-t001:** General characteristics of the three soils used for the acclimation of willow plantlets. Values are presented as mean ± standard deviation.

		Soil Type		
		Peat	Forest	Crop
pH H_2_O		5.4 ± 0.06	4.5 ± 0.05	5.2 ± 0.11
pH KCl		4.9 ± 0.01	3.5 ± 0.01	4.0 ± 0.01
% total C		43.2 ± 0.9	12.6 ± 0.8	2.5 ± 0.2
% total N		1.2 ± 0.02	0.6 ± 0.02	0.2 ± 0.00
C/N		38	21	15
Total P		505 ± 30	452 ± 31	591 ± 31
Available P ^#^	Inorganic P	298.4 ± 13.5	6.6 ± 2.3	52.9 ± 0.7
	Organic P	26.5 ± 16.3	36.6 ± 2.3	25.8 ± 1.0
Inorganic N ^#^	N-NH_4_^+^	346.9 ± 2.9	33.5 ± 1.4	19.3 ± 1.4
	N-NO_3_^−^	257.6 ± 6.4	9.1 ± 3.2	9.1 ± 3.2
Soluble C ^#^ (hot water)		6621 ± 107	4256 ± 234	899 ± 39

^#^ Values expressed in mg kg^−1^.

**Table 2 plants-12-00132-t002:** Effect of the type of soil, the time of sampling and the presence of plants on enzymatic activities of peat, forest and crop soil used for willow acclimation. Values are presented as mean ± standard deviation of 5 samples per soil for the 2 controls and 20 samples per soil and sucrose treatment for the soils with plants. Control t0: control without plants at time 0 (when the plants were transferred to different soils); Control 6w: control without plants after 6 weeks; Plant S0/S3: samples with plants micropropagated with 3% sucrose or without sucrose after 6 weeks of planting. Bold numbers indicate the soil with highest enzymatic activity for each enzyme.

Enzyme		Peat	Forest	Crop
**Urease** (µmol NH_3_ g^−1^ h^−1^)	Control t 0	4.46 ± 0.27	**11.78 ± 0.80**	5.40 ± 0.07
	Control t 6 weeks	2.83 ± 0.05	**17.15 ± 1.26**	9.36 ± 0.87
	Plants S3	5.33 ± 0.47	**19.75 ± 2.33**	10.95 ± 0.87
	Plants S0	5.39 ± 0.65	**17.23 ± 1.24**	11.10 ± 5.12
**Dehydrogenase** (µmol INTF g^−1^ h^−1^)	Control t 0	**0.54 ± 0.02**	0.49 ± 0.01	0.38 ± 0.03
	Control t 6 weeks	**0.52 ± 0.01**	0.43 ± 0.01	0.34 ± 0.02
	Plants S3	**0.63 ± 0.06**	0.45 ± 0.04	0.33 ± 0.01
	Plants S0	**0.75 ± 0.04**	0.48 ± 0.03	0.33 ± 0.00
**Acid phosphomonoesterase** (µmol PNP g^−1^ h^−1^)	Control t 0	5.04 ± 0.79	**6.96 ± 0.08**	2.93 ± 0.07
	Control t 6 weeks	2.75 ± 0.10	**7.88 ± 0.37**	2.51 ± 0.18
	Plants S3	3.12 ± 0.64	**7.24 ± 0.90**	2.31 ± 0.14
	Plants S0	3.46 ± 0.44	**8.19 ± 0.30**	2.44 ± 0.14
**ß-glucosidase** (µmol PNG g^−1^ h^−1^)	Control t 0	**1.53 ± 0.06**	1.86 ± 0.09	0.43 ± 0.03
	Control t 6 weeks	**4.18 ± 0.11**	1.53 ± 0.08	0.55 ± 0.02
	Plants S3	**3.97 ± 0.24**	1.59 ± 0.13	0.53 ± 0.06
	Plants S0	**4.25 ± 0.19**	1.60 ± 0.08	0.51 ± 0.03
**Invertase** (µmol Glu g^−1^ h^−1^)	Control t 0	1.68 ± 0.09	**11.19 ± 0.33**	4.01 ± 0.01
	Control t 6 weeks	1.86 ± 0.12	**8.32 ± 0.30**	3.51 ± 0.36
	Plants S3	2.54 ± 0.28	**8.61 ± 0.63**	3.68 ± 0.35
	Plants S0	2.18 ± 0.51	**8.07 ± 0.53**	3.81 ± 0.10
**Arylsulfatase** (µmol PNS g^−1^ h^−1^)	Control t 0	0.09 ± 0.06	**0.35 ± 0.02**	0.16 ± 0.01
	Control t 6 weeks	0.04 ± 0.00	**0.46 ± 0.02**	0.17 ± 0.01
	Plants S3	0.09 ± 0.04	**0.42 ± 0.06**	0.19 ± 0.01
	Plants S0	0.09 ± 0.02	**0.43 ± 0.02**	0.19 ± 0.01

## Data Availability

Not applicable.

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
