# Peer review of "Effect of Soil Type and In Vitro Proliferation Conditions on Acclimation and Growth of Willow Shoots Micropropagated in Continuous Immersion Bioreactors"

_plants, 2022, doi:10.3390/plants12010132_

Round 1

Reviewer 1 Report

The manuscript title “Effect of Soil Type and In Vitro Proliferation Conditions on Acclimation of Willow Shoots Micropropagated in Continuous Immersion Bioreactors” is an excellent study and have scientific worth. This MS need minor improvements. My comments for authors are as follows:

1-      Please add the horizontal and vertical axis title in all graphs.

2-      The author performed ANOVA, where is ANOVA table? I suggest authors to add ANOVA table in the MS.

3-      Figures 3, 4 and 8: why author write Aa, Bb in above the bar columns, just write a, b is also ok?.

4-      4.2 and 4.3 headings are same???

5-      Pleas add a conclusion section in the MS.

Author Response

The authors acknowledge the comments and suggestion of reviewer 1 for improving the manuscript, and have performed the required changes. Here are our answers

My comments for authors are as follows:

1-      Please add the horizontal and vertical axis title in all graphs.

Done

2-      The author performed ANOVA, where is ANOVA table? I suggest authors to add ANOVA table in the MS.

ANOVA tables have been included as supplementary material.

3-      Figures 3, 4 and 8: why author write Aa, Bb in above the bar columns, just write a, b is also ok?

In those Figures we performed two-way ANOVA. For this reason we used two types of letters, to facilitate the understanding of the contribution of the two factors under study to the variance between groups. Below we show some examples of the use of uppercase and lowercase letters for two-way ANOVA results in recent papers published in high quality journals as MDPI and Frontiers.

Fritz, V.; Tereucán, G.; Santander, C.; Contreras, B.; Cornejo, P.; Ferreira, P.A.A.; Ruiz, A. Effect of Inoculation with Arbuscular Mycorrhizal Fungi and Fungicide Application on the Secondary Metabolism of Solanum tuberosum Leaves. Plants 2022, 11, 278. https://doi.org/10.3390/plants11030278

Cirillo, A.; Magri, A.; Scognamiglio, M.; D’Abrosca, B.; Fiorentino, A.; Petriccione, M.; Di Vaio, C. Evaluation of Morphological, Qualitative, and Metabolomic Traits during Fruit Ripening in Pomegranate (Punica granatum L.). Horticulturae 2022, 8, 384. https://doi.org/10.3390/horticulturae8050384

Lin Weihu, Kuang Yu, Wang Jianjun, Duan Dongdong, Xu Wenbo, Tian Pei, Nzabanita Clement, Wang Meining, Li Miaomiao, Ma Bihua. Effects of Seasonal Variation on the Alkaloids of Different Ecotypes of Epichloë Endophyte-Festuca sinensis Associations.  Frontiers in Microbiology 10, 2019. https://www.frontiersin.org/articles/10.3389/fmicb.2019.01695

4-      4.2 and 4.3 headings are same???

The authors acknowledge the reviewer comment. It was a typographic mistake and now it has been amended.

5-      Pleas add a conclusion section in the MS.

Done

Reviewer 2 Report

Dear Authors,

Thank you for submitting your manuscript to the Plants MDPI journal.  I have a few suggestions to improve your manuscript.

Title:

“Effect of Soil Type and In Vitro Proliferation Conditions on Acclimation of Willow Shoots Micropropagated in Continuous Immersion Bioreactors.”

In chapter 4.2. Acclimation and Effect of the Type of Soil you show that:

“Rooted shoots obtained by CIS under photomixotrophic and photoautotrophic conditions were selected for evaluating different soils during the acclimation step. Uniform shoots about 70 mm height were chosen, and their roots were uniformly cut to 30 mm length from the base of the shoots and planted in commercial peat for a first acclimation step in pots of 300 ml. The pots were transferred to a controlled environmental chamber (Fitotron SGC066, Sanyo Gallencamp PLC) with a photoperiod of 16 h light: 8 h dark, a 487 photosynthetic photon flux of 240-250 μmol m-2s-1, a temperature of 25 ËšC (day) and 20 ËšC  (night), and relative humidity of 85%. The plantlets were sprayed with water daily, watered twice a week and measured at 2-week intervals. Four weeks after transplanting the aerial and root sections were measured and roots were carefully washed to eliminate the rests of peat before being transferred to new pots (total volume 1,3 L, substrate 1 L) in the greenhouse. “

I consider that the acclimatization took place during this period. After planting in the three types of soil in the greenhouse, you followed the growth of the plants.

Therefore, the title should be changed as follows

“Effect of Soil Type and In Vitro Proliferation Conditions on Growth of Willow Shoots Micropropagated in Continuous Immersion Bioreactors.”

If you agree with the change you should reinterpret the discussion and conclusion.

Introduction

Line 56-58 represents the aim of the study and should be moved to the last paragraph of the introduction

I suggest you improve the introduction with a paragraph about the willow acclimation

Material and Methods

The chapter 4.1. is a bit confusing and should be restructured. A structure guideline might be: the first experiment..... the second experiment

Line 481 - 4.2 Acclimation and Effect of the Type of Soil … and line 504 - 4.3 Acclimation and Effect of the Type of Soil ….are identical?

Results

Line 121-124 - I think this paragraph should be moved to the discussion chapter

Author Response

The authors acknowledge the comments and suggestion of reviewer 2 for improving the manuscript, and have performed the required changes.

Dear Authors,

Thank you for submitting your manuscript to the Plants MDPI journal.  I have a few suggestions to improve your manuscript.

Title:

“Effect of Soil Type and In Vitro Proliferation Conditions on Acclimation of Willow Shoots Micropropagated in Continuous Immersion Bioreactors.”

In chapter 4.2. Acclimation and Effect of the Type of Soil you show that:

“Rooted shoots obtained by CIS under photomixotrophic and photoautotrophic conditions were selected for evaluating different soils during the acclimation step. Uniform shoots about 70 mm height were chosen, and their roots were uniformly cut to 30 mm length from the base of the shoots and planted in commercial peat for a first acclimation step in pots of 300 ml. The pots were transferred to a controlled environmental chamber (Fitotron SGC066, Sanyo Gallencamp PLC) with a photoperiod of 16 h light: 8 h dark, a 487 photosynthetic photon flux of 240-250 μmol m-2s-1, a temperature of 25 ËšC (day) and 20 ËšC  (night), and relative humidity of 85%. The plantlets were sprayed with water daily, watered twice a week and measured at 2-week intervals. Four weeks after transplanting the aerial and root sections were measured and roots were carefully washed to eliminate the rests of peat before being transferred to new pots (total volume 1,3 L, substrate 1 L) in the greenhouse. “

I consider that the acclimatization took place during this period. After planting in the three types of soil in the greenhouse, you followed the growth of the plants.

Therefore, the title should be changed as follows: “Effect of Soil Type and In Vitro Proliferation Conditions on Growth of Willow Shoots Micropropagated in Continuous Immersion Bioreactors.”

If you agree with the change you should reinterpret the discussion and conclusion.

We understand the point of view of the reviewer 2. Without having carried out physiological analysis that monitor changes on photosynthesis, water translocation, leaf structure, stomas, etc. of the plantlets during the stage in the phytotron and the greenhouse it is difficult to say when the acclimation has been completed. For this reason we have changed the title and the manuscript according to the reviewer suggestion.

Introduction

Line 56-58 represents the aim of the study and should be moved to the last paragraph of the introduction

Done

I suggest you improve the introduction with a paragraph about the willow acclimation

Done

Material and Methods

The chapter 4.1. is a bit confusing and should be restructured. A structure guideline might be: the first experiment..... the second experiment

Done

Line 481 - 4.2 Acclimation and Effect of the Type of Soil … and line 504 - 4.3 Acclimation and Effect of the Type of Soil ….are identical?

We acknowledge the reviewer comment. It was a typographic mistake and now it has been amended.

Results: Line 121-124 - I think this paragraph should be moved to the discussion chapter

We acknowledge the reviewer comment. The experiment belongs to the present study, but it had a typographic mistake and it was misleading. We have rewritten the sentence to clarify the meaning.

Round 2

Reviewer 2 Report

Most problems in the manuscript have been revised and I suggest accept this study in present form.